# Explainable and Efficient Editing for Large Language Models

## Abstract

Large Language Models (LLMs) possess remarkable capabilities in storing and retrieving vast factual knowledge but often retain outdated or incorrect information from web corpora. While full retraining is costly, locate-and-edit model editing methods offer an feasible alternative. Current methods typically follow a two-stage paradigm: (1) identifying critical layers for knowledge storage and (2) updating their parameters to store new knowledge. However, both of these two phases have their inherent limitations. In stage 1, layers identification is independent of the to-be-updated knowledge, ignoring the varying storage patterns of different knowledge types. Meanwhile, Stage 2 suffers from high computational overhead due to independent gradient descent for each piece of knowledge. To solve these, we propose an **E**xplainable and effi**C**ient model **E**diting method, termed **ECE**. Specifically, in Stage 1, ECE integrates the concept of LLMs explainability into the editing process, enabling the adaptive identification of the crucial neurons based on the input knowledge. In Stage 2, ECE clusters similar knowledge based on the explanation results, allowing batch optimization in a single gradient step, significantly reducing time consumption without sacrificing effectiveness. Extensive experiments demonstrate that ECE can achieve superior performance while delivering a 3.27× speedup in editing efficiency, showcasing the potential of explainability-driven editing methods for LLMs.

## Keywords

Large Language Models, Knowledge Editing, Model Explainability

## 1 Introduction

Large Language Models (LLMs) have recently demonstrated remarkable capabilities in storing vast amounts of factual knowledge and retrieving it effectively during inference [8, 41, 54]. The knowledge in LLMs primarily stems from the extensive training data, particularly web corpora. However, these corpora often contain inaccuracies and outdated information that LLMs may inadvertently store, necessitating targeted modifications to correct these knowledge bases [19, 48]. While retraining the entire LLM is a direct solution, it is resource-intensive, both in terms of time and computational cost [44, 47]. As an efficient alternative, locate-and-edit model editing methods have emerged for updating specific knowledge [9, 37, 38]. These methods generally follow a two-stage paradigm: (1) given an LLM, identifying the layers most critical to knowledge storage by causal tracing; (2) given a new piece of

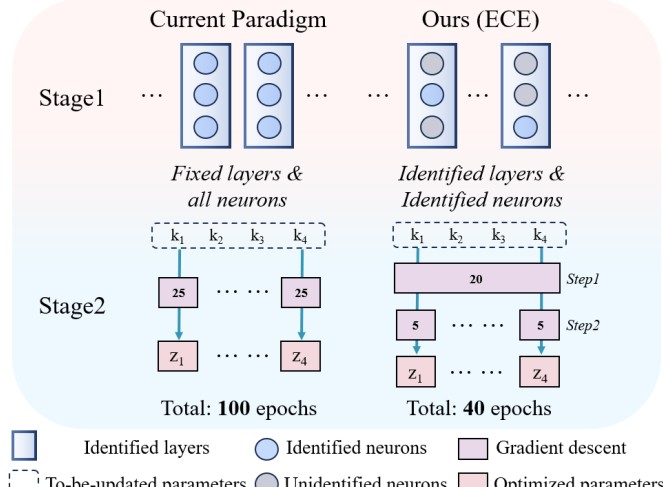

**Figure 1: Overview of current methods and ours at Stage 1 and 2 in sequential model editing.**

knowledge, computing the optimal output of the identified layers to ensure correct responses. The optimal output is then employed to update the critical layers' parameters, allowing for knowledge updates by adjusting only a small subset of model parameters [65].

While this two-stage paradigm is effective, both stages present inherent issues [21, 24, 34, 36]. Specifically,

- As shown in the top half of Figure 1, in Stage 1 (*i.e.* critical layer identification), the to-be-updated layers and parameters remain fixed regardless of the type of new knowledge. However, recent studies on LLMs explainability suggest that different types of knowledge are stored in distinct layers and neurons [49, 59]. Current methods in Stage 1 fail to leverage this explainable knowledge-neuron correspondence, resulting in the inability to adaptively identify layers based on the input knowledge, which leads to sub-optimal editing performance.

- As depicted in the bottom half of Figure 1, in Stage 2 (*i.e.* parameter update), the process is computationally expensive, as the optimal outputs of the critical layers must be calculated independently for each knowledge instance. In practice, the number of knowledge updates could exceed tens of thousands, imposing significant efficiency constraints. Worse still, in lifelong editing scenarios (*i.e.*, continuous updating the same LLM [28, 67]), each update has to modify all key layers and neurons identified in Stage 1 [7, 18], significantly increasing time consumption.

Thus, a key question arises: *Can we design an explainable and efficient editing method that adaptively identifies key neurons in Stage 1 and streamlines parameter updates in Stage 2?*

To answer this question, we propose an **E**xplainable and effi**C**ient sequential **E**diting method, called **ECE**. Specifically, in Stage 1, ECE integrates the concept of LLMs explainability [48, 64, 71] into the editing process, enabling the adaptive identification of the most

relevant layers and neurons based on the input knowledge [71]. This identification is inspired by the advanced attribution methods in LLMs explainability (*i.e.*, activation-based [11], weight-based [60], and residual-flow-based methods [42, 49]), enabling ECE to focus on the most critical parameters solely. By isolating neurons unrelated to the updated knowledge, ECE safeguards the integrity of other knowledge stored within the LLM. In a nutshell, ECE introduces LLMs explainability to improve editing performance.

Furthermore, the explainability introduced in Stage 1 serves as a foundation for accelerating Stage 2 (*i.e.*, parameter update). This acceleration is manifested in two key aspects: (1) Unlike current updates (*i.e.*, updating all parameters in every key layer), ECE performs layer- and neuron-wise updates (*i.e.*, updating only a limited set of parameters within the selected layers, which are identified by the attribution approaches), significantly reducing the overall parameter volume. (2) Current research has verified that knowledge instances of similar types often exhibit consistent distributions of key neurons and optimal outputs across key layers [10, 20]. Leveraging this insight, ECE employs advanced clustering algorithms [42, 49] to group type-similar knowledge based on the distribution of key neurons. This allows us to compute the optimal outputs for these knowledge instances simultaneously in a single gradient descent step, drastically reducing the time-consuming gradient descent process. These strategies collectively accelerate the parameter update process and enhance the efficiency of model editing.

We conduct extensive qualitative and quantitative experiments on GPT2-XL (1.5B) [46], GPT-J (6B) [58], and Llama-3 (8B) [16]. Results across multiple datasets demonstrate that, compared to the baselines (*e.g.*, Fine-tuning [53], MEND [39], ROME [37], and MEMIT [38]), ECE significantly outperforms in editing effectiveness across several metrics, including efficacy, generalization, specificity, fluency, and consistency. Moreover, ECE achieves an average speedup of 3.27× in editing efficiency for sequence editing. These findings confirm that incorporating LLM explainability to streamline the editing process can lead to improvements in both effectiveness and efficiency.

Our key contributions are summarized as follows:

- We systematically analyze the inherent issues in current locate-and-edit editing methods, specifically the lack of explainability and inefficiency during the critical layer identification and parameter update phases.
- We propose a novel sequential editing method, termed ECE. By integrating attribution methods from LLM explainability, ECE adaptively identifies and updates neurons within LLMs, achieving improvements in both effectiveness and efficiency.
- Experiments across multiple LLMs demonstrate that ECE outperforms leading editing methods across five general evaluation metrics and two commonly used datasets.

## 2 Preliminary

Based on prior works [9, 37, 38], model editing aims to modify an initial base model $f_\theta$ (where $\theta$ represents the model's parameters) into an edited version $f_{\theta'}$. The objective is to adjust the model's responses to a specified set of knowledge instance, while preserving its performance on all other knowledge instances [1, 32]. The intended edit descriptor is denoted as $\{(x_i^e, y_i^e)\}_{i \in [1,N]}$, where

$f_\theta(x_i^e) \neq y_i^e$, and $N$ represents the total number of editing instances. This set of instances forms the editing scope $I_{edit}$, while $O_{edit}$ represents the instances outside the editing scope. Formally, a successful edit can be expressed as:

$$f_{\theta'}(x_i) = \begin{cases} y_i^e, & \text{if } x_i \in I_{edit}, \\ f_\theta(x_i), & \text{if } x_i \in O_{edit}. \end{cases} \quad (1)$$

Sequential model editing [36, 70] refers to the process of continuously refining a pre-trained model, $f_{\theta_0}$, through a series of updates, where each update incorporates modifications or corrections to adjust the model's outputs [65, 69]. This process is expressed as:

$$\theta' \leftarrow \arg\min_\theta \left( \sum_{s=0}^{S} \sum_{i=s \times n}^{(s+1) \times n} \|f_\theta(x_i^e) - y_i^e\| \right), \quad (2)$$

where $n$ represents the batch size, and $S$ represents the sequential editing step.

In practice, each update involves introducing a set of factual triples in the form of $(s, r, o)$, where $s$ represents the subject, $r$ the relation, and $o$ the object (*e.g.*, $s$="The largest ocean", $r$="is", $o$="Pacific Ocean"). After the $t$-th edit, the updated model $f_{\theta_t}$, built on its predecessor $f_{\theta_{t-1}}$, is optimized to generate precise target outputs for the corresponding inputs $\mathbb{D}_{edit_t}$, while preserving accuracy on inputs outside the current editing scope. Adopting methods from ROME [37] and MEMIT [38], we conceptualize the feed-forward network (FFN) layer of a Transformer [55] as a linear associative memory. This approach effectively utilizes linear mappings within the FFN to serve as key-value pairs for information retrieval [2, 30]. Our objective is to adjust the output of the LLM such that the input $(s_i, r_i)$ produces the output $o_i$.

The process begins by identifying the activation output from the last subject token $S$ at the $l$-th FFN layer, which serves as the key $k_i^l$. These keys are computed from the input weights $W_{in}^l$ and are processed through the output weights $W_{out}^l$ to generate the corresponding values $v_i^l$. This setup allows us to capture the LLM's inherent $n$ knowledge pairs, associating input keys $K_0 = [k_1 \,|\, k_2 \,|\, \ldots \,|\, k_n]$ with corresponding values $V_0 = [v_1 \,|\, v_2 \,|\, \ldots \,|\, v_n]$. We aim to integrate $u$ additional key-value pairs associated with new knowledge, denoted as $K_1 = [k_{n+1} \,|\, k_{n+2} \,|\, \ldots \,|\, k_{n+u}]$ and $V_1 = [v_{n+1} \,|\, v_{n+2} \,|\, \ldots \,|\, v_{n+u}]$, while preserving the original associations. The values $V_1$ are optimized through gradient descent to maximize the probability of the target token outputs, as detailed in MEMIT [38].

The optimization framework is defined as:

$$\Delta = \arg\min_{\hat{\Delta}} \left( \left\| (W + \hat{\Delta})K_1 - V_1 \right\|^2 + \left\| (W + \hat{\Delta})K_0 - V_0 \right\|^2 \right), \quad (3)$$

where $W$ represents the output weights of the target FFN layer and $\Delta$ denotes the required weight updates. The knowledge retention can be expressed as $WK_0 = V_0$. Utilizing the least squares method [31], the optimal weight update $\Delta$ is calculated as follows:

$$\Delta = RK_1^T \left( K_0 K_0^T + K_1 K_1^T \right)^{-1}, \quad (4)$$

where $R = V_1 - WK_1$. Here, the matrix $K_0 K_0^T$ can be approximated by $\lambda \mathbb{E}[kk^T]$, a covariance statistic without centering, derived from an empirical dataset of vector inputs to the layer.

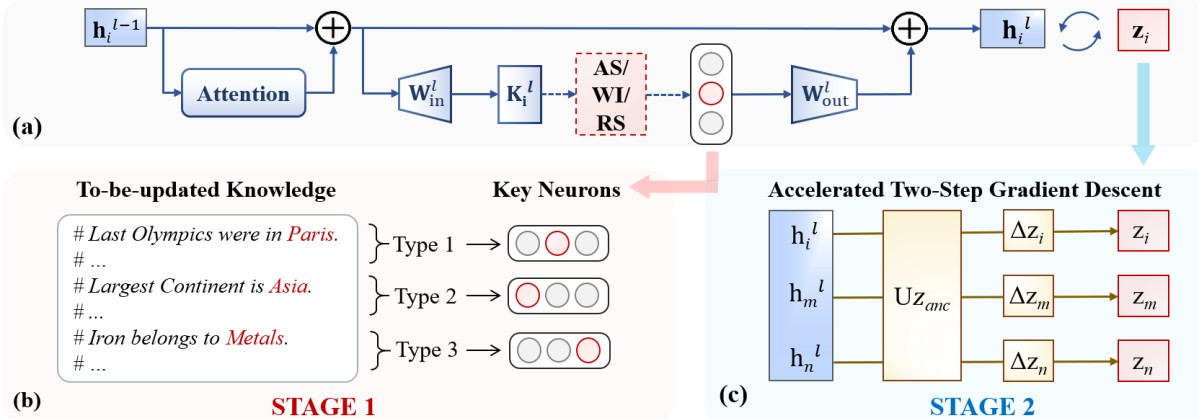

**Figure 2: Overview of ECE. (a) The neuron-wise identification approach is based on attribution methods using Activation Score, Weight Importance, or Residual Sensitivity.** $h$ **denotes hidden state,** $z$ **denotes optimal representation, and the red color highlights the identified neurons that play a critical role in storing or processing specific knowledge. (b) The clustering method applies to neurons corresponding to knowledge instances. (c) The acceleration is achieved by the two-step gradient descent method based on the clustering results. U and** $\Delta$ **in yellow represent the step-1 and step-2 gradient descent.**

## 3 Methodology

In this section, we detail how ECE achieves simultaneous improvements in efficiency and effectiveness through the incorporation of explainability. Specifically, in Section 3.1, we present neuron-wise identification that employs mature attribution algorithms to locate fine-grained layers and neurons. In Section 3.2, we demonstrate how the attribution results accelerate the learning of key matrices, thereby significantly enhancing editing efficiency. Finally, Section 3.3 outlines the parameter update process based on the findings from Section 3.1 and 3.2.

### 3.1 Neuron Identification

We first revisit the process of identifying parameters for updating in current model editing approaches. Existing methods [37, 38] primarily use causal tracing [37] to pinpoint layers with the highest causal effect, assuming these layers store key knowledge in the LLM. However, this approach has two major limitations:

- **Independence from Specific Knowledge:** Once key layers are identified, all neurons within them are treated equally during editing, overlooking the fact that different types of knowledge are encoded in distinct patterns [25]. Each neuron plays a unique role — some are crucial for specific information, while others may be less relevant or even inactive for certain tasks;
- **Overlooking Neurons Outside Key Layers:** Focusing solely on selected layers risks missing important neurons distributed across other layers that also contribute to knowledge storage and retrieval. Moreover, research about "dead neurons problem" [17, 56] points that inactive or "dead" neurons consume capacity without contributing meaningfully to the model's output. Editing both inactive and critical neurons indiscriminately can reduce the precision of updates and introduce unnecessary disruptions to the model's balance.

These limitations highlight the need for more precise and efficient approaches to knowledge editing beyond layer-level modifications.

To solve these, an intuitive optimization is to adaptively identify key layers and neurons for editing based on to-be-updated knowledge. This layer- and neuron-wise editing approach allows for more precise modifications while minimizing disruptions to other stored knowledge. Drawing inspiration from well-established neuron attribution methods in LLM explainability [43, 60], we employ three attribution methods to rank neuron relevance according to the to-be-updated knowledge as follows:

- **Activation Score (AS)** [42] ranks neurons directly by the magnitude of their activation values during inference, identifying those that are highly active in processing specific inputs. Formally:

$$AS_i = |a_i(x_j)|, \tag{5}$$

where $a_i(x_j)$ denotes the activation value of neuron $i$ for knowledge instance $x_j$.

- **Weight Importance (WI)** [49] evaluates neurons based on the weights involved in transmitting information between neurons, emphasizing the significance of neurons with stronger internal connections. The importance score is defined as:

$$WI_i = |W_{ij}|, \tag{6}$$

where $W_{ij}$ denotes the weight between neuron $i$ and neuron $j$.

- **Residual Sensitivity (RS)** [50] assesses neurons by their contribution to the final output through the residual stream. The importance score is defined as:

$$RS_i = a_k^l (W_{out}^l)_k, \tag{7}$$

where $a_k^l$ is the activation value of neuron $k$ in layer $l$, $(W_{out}^l)_k$ denotes the output weight for neuron $k$ in layer $l$.

Building on prior work that suggests that neurons within Feed-Forward Networks (FFNs) contain significant amounts of specific factual information [11, 42, 49, 59], we prioritize the optimization of selected neurons rather than modifying the entire parameter space. For each knowledge instance $(s_i, r_i, o_i)$, we compute the score values from one of the three methods above and obtain scores $Q_i$

for the neurons. Using a descending sorting method, we then rank and group the neurons with the highest scores together. We then select a subset of neurons by identifying those whose cumulative score exceeds a predetermined fraction $p$ of the total score:

$$\mathcal{I} = \arg \min_{\mathcal{I} \subseteq \{1,\dots,N\}} |\mathcal{I}| \quad \text{s.t.} \quad \sum_{j \in \mathcal{I}} \mathbf{Q}_{ij} \geq p \cdot \sum_{j=1}^{N} \mathbf{Q}_{ij}, \qquad (8)$$

where $\mathbf{Q}_{ij}$ represents the score of the $j$-th neuron, and $\mathcal{I}$ is the selected set of neuron indices. This identification mechanism enables targeted edits, ensuring efficiency and precision in the parameter update process.

Given that the model may need to edit multiple knowledge facts in parallel (*i.e.*, in a batch), which may correspond to different neurons across the network, we aggregate neuron scores across all batch samples. This allows for a unified neuron identification based on cumulative influence contribution, which streamlines the model's response to various edits.

## 3.2 Accelerated Learning of Key Matrices

After identifying the to-be-updated parameters $\mathcal{I}$, the next step is to obtain the optimal representations of the relevant layers post-editing, *i.e.*, the value matrix $V_1$ for the to-be-updated knowledge as outlined in Section 2. Note that this step serves as a key driver of ECE's acceleration by leveraging the attribution results from the above Section. Next, we will detail how ECE achieves simultaneous improvements in efficiency and effectiveness through two progressive steps: knowledge clustering and two-step gradient descent.

*3.2.1 Knowledge Clustering.* Different types of knowledge inherently possess varying textual attributes such as geographical, demographic, and temporal concepts, which implies that their key information is stored in different regions within the model [42, 49]. Hence, we propose a clustering approach to pre-classify knowledge, thereby avoiding conflicts that may arise due to the distinct characteristics of the knowledge being edited. Specifically, we represent each knowledge instance $x_i$ and its corresponding set of identified neurons as a key-value pair. By applying a k-means clustering algorithm based on Jaccard similarity, we aggregate knowledge instances with similar neuron identifications into clusters. In our clustering approach, the objective is to minimize the Jaccard distance between the data points and their respective cluster centroids, formulated as:

$$\arg \min_{S} \sum_{i=1}^{k} \sum_{x_j \in S_i} d_J(x_j, c_i), \qquad (9)$$

where $S_i$ represents the $i$-th cluster, $x_j$ denotes a data point in cluster $S_i$, and $c_i$ is the centroid of cluster $S_i$. By minimizing this objective, we ensure that the knowledge within each cluster exhibits high internal similarity.

We treat each resulting cluster as a smaller batch and subsequently use the sample closest to the center of each cluster as an anchor sample. The anchor sample for each cluster is defined as the sample with the smallest sum of Jaccard distances to all other samples in the cluster. This criterion ensures that the selected anchor is the most representative data point of its cluster, which is

formulated as follows:

$$x_{\text{anc}} = \arg \min_{x_j \in S_i} \sum_{x_k \in S_i} d_J(x_j, x_k). \qquad (10)$$

This approach allows us to perform subsequent editing tasks in a more fine-grained manner, tailored to the specific attributes of each knowledge category.

*3.2.2 Two-step Gradient Descent.* To enhance computational efficiency in the sequential batch editing process, we introduce a two-step gradient descent approach applied to the clusters identified in the previous step. For each cluster, the gradient descent is divided into a common phase and an instance-specific phase, allowing us to maximize shared information while maintaining unique adjustments for each instance within the cluster. This approach significantly reduces redundant calculations and accelerates the model adaptation process.

In the first phase, we perform a unified gradient descent over the entire cluster, conducting 20 epochs of shared updates from a total of 25 epochs for the whole process. In this modified shared gradient descent, all instances within the cluster share the update vector $z_{\text{anc}}$ associated with the anchor sample $x_{\text{anc}}$ of the cluster for the first 20 epochs. This shared step captures common patterns by optimizing parameters based on the anchor sample, which is broadly representative of the entire cluster, thereby reducing the number of repetitive updates. Let $C_u$ represent the $u$-th cluster, and let $h_{\text{anc}}^L$ denote the update vector for the anchor sample within the cluster. By minimizing the average loss based on the anchor sample $x_{\text{anc}}$ and its corresponding edit target $y_{\text{anc}}^e$, we calculate the unified update $\mathcal{U}z_u$ for the entire cluster as follows:

$$\mathcal{U}z_u = \arg \min_{\delta} -\log \mathbb{P}_G(h_{\text{anc}}^L + \delta)(y_{\text{anc}}^e \mid x_{\text{anc}}), \qquad (11)$$

This unified update step captures the general characteristics of the cluster by leveraging the anchor sample, thereby setting a common foundation for the individual updates that follow.

In the second phase, we refine this update by performing an additional five epochs of gradient descent tailored to each instance within the cluster. This step fine-tunes the model on unique variations and specific details, accommodating the individual characteristics and ensuring that the final updates are well-adapted to each instance. The optimization objective is as follows:

$$\delta_i^* = \arg \min_{\delta_i} -\log \mathbb{P}_G(h_i^L + \mathcal{U}z_u + \delta_i)(y_i^e \mid x_i), i \in C_u, \qquad (12)$$

where $\delta_i^*$ is the instance-specific adjustment derived by minimizing the residual error after applying the shared update $\mathcal{U}z_k$. Thus, the two-step update $z_i$ for a specific instance $i$ in cluster $C_k$ can be integrated together as:

$$z_i = h_i^L + \mathcal{U}z_u + \delta_i^*. \qquad (13)$$

This step preserves individual differences by fine-tuning the shared update to better align with the specific characteristics and requirements of each instance.

By separating the optimization into these two steps, we achieve both efficiency and adaptability. The unified 20-epoch update captures the core features shared among instances within a cluster,

while the five-epoch instance-specific phase ensures that each instance receives the necessary unique adjustments. This design reduces the overall computation required by avoiding repetitive updates for common features, which is particularly valuable in large-scale sequential batch editing scenarios. Consequently, this method provides a balanced approach that maintains the specificity of individual updates while optimizing shared computations, leading to faster convergence and lower computational costs in comparison to traditional instance-by-instance gradient descent methods.

Following the explainable neuron identification approach, we have identified the most influential neurons responsible for encoding the relevant knowledge. By focusing our updates solely on these critical neurons, rather than performing a full update across all neurons, we achieve a more efficient and targeted editing process. This selective update strategy allows us to concentrate computational resources on the neurons that most directly impact the model's output, effectively reducing unnecessary overhead associated with updating less significant parts of the network. Unlike traditional full-scale updates that modify the entire layer or model parameters indiscriminately, our approach not only accelerates the editing process but also minimizes potential disruptions to the model's stability and integrity.

## 3.3 Parameter Updates

After identifying the to-be-updated parameters $\mathcal{I}$ in Section 3.1 and the value matrix $V_1$ in Section 3.2, we arrive at the final step: performing the parameter update on $\mathbf{W}_{out}$. Let $\hat{\mathbf{W}}$ and $\hat{\Delta}$ denote the submatrices of $\mathbf{W}$ and the update $\Delta$, respectively, formed by selecting rows indexed by $\mathcal{I}$. Our objective is to optimize the updated parameters for each neuron set by minimizing the squared error between the model's output and the target knowledge representations:

$$\hat{\Delta} = \arg\min_{\hat{\Delta}} \left( \left\| (W + \hat{\Delta})K_1 - V_1 \right\|^2 + \left\| (W + \hat{\Delta})K_0 - V_0 \right\|^2 \right), \quad (14)$$

where $\hat{K}_0$ and $\hat{K}_1$ are two submatrix formed from $\mathbf{Q}_0$ and $\mathbf{Q}_1$ by indexing the columns corresponding to $\mathcal{I}$. This formulation ensures that the model retains previously learned knowledge (through $K_0$) while incorporating new edits (through $K_1$).

Following MEMIT [38], we derive the solution for Eqn. 15 using the method of minimal squared error as:

$$\hat{\Delta}^* = \hat{\mathbf{R}}\hat{\mathbf{K}}_1^T\hat{\mathbf{C}}^{-1}, \quad (15)$$

where $\hat{\mathbf{R}} = \mathbf{V}_1 - \hat{\mathbf{W}}\hat{\mathbf{K}}_1$ and $\hat{\mathbf{C}} = \hat{\mathbf{K}}_0\hat{\mathbf{K}}_0^T + \hat{\mathbf{K}}_1\hat{\mathbf{K}}_1^T$. In order to maintain continuity, we approximate $\mathbf{K}_0\mathbf{K}_0^T$ with $\lambda\mathbb{E}\left[\mathbf{kk}^T\right]$, where $\lambda$ is a hyperparameter balancing the retention of prior knowledge with the integration of new edits. The submatrix $\hat{\mathbf{K}}_0\hat{\mathbf{K}}_0^T$ is then derived from $\mathbf{K}_0\mathbf{K}_0^T$ by indexing only the identified neurons. Additionally, as each editing round progresses, newly edited knowledge becomes the reference knowledge for subsequent rounds, which requires updating $\mathbf{K}_0\mathbf{K}_0^T$ after each iteration.

## 4 Experiments

We conduct experiments to demonstrate the effectiveness of our model editing method. The experiments aim to address the following research questions:

- **RQ1:** How does ECE's performance on sequential model editing tasks measure up against existing methods?
- **RQ2:** What is the impact of different parameter settings on the performance and stability of sequential model editing?
- **RQ3:** How much efficiency improvement can ECE achieve in comparison to existing editing techniques?
- **RQ4:** Can LLMs preserve the original general abilities after extensive sequential edits?

## 4.1 Experimental Settings

**Datasets & Evaluation Metrics.** To evaluate the effectiveness of our method, we utilize two datasets: Counterfact [37] and ZsRE [33]. For the Counterfact dataset, we utilize five evaluation metrics as defined in previous studies [37, 38]: **Efficacy** (efficiency success), **Generalization** (paraphrase success), **Specificity** (neighborhood success), **Fluency** (generation entropy), and **Consistency** (reference score). For the ZsRE dataset, we apply three evaluation metrics, also defined in previous work [37–39]: **Efficacy**, **Generalization**, and **Specificity**. For more details, see Appendix C.

**Baselines:** For baseline comparisons, we consider several model editing approaches across different categories. (1) Fine-tuning based: **FT-L** [53] directly fine-tunes a single layer's feed-forward network (FFN), and **FT-M** [73] is a variation of FT-L with a different loss computation; (2) Locate-and-edit: **ROME** [37] which identifies critical neuron activations within middle-layer feed-forward modules that influence factual prediction and **MEMIT** [38] treats the transformer's feed-forward layer as a linear associative memory and applies minimum square error optimization to introduce new key-value associations; (3) Meta-learning based: **MEND** [39] uses a hyper-network to transform gradients obtained via standard fine-tuning; (4) Memory-based: **SERAC** [40] which employs an external cache to store explicit edits.

**Implementation Details:** Our comparative analysis evaluates the performance of various editing methods on three autoregressive language models, GPT2-XL (1.5B) [46], GPT-J (6B) [58] and Llama3 (8B) [16] . In addition to covering two widely adopted GPT models built on the classic transformer architecture, we include Llama3, one of the most powerful models available in the current open-source landscape. Further details of the implementation are provided in the Appendix D.

## 4.2 Editing Performance (RQ1)

In this subsection, we present a detailed comparison of ECE against other established methods for the sequential model editing task, conducted using GPT2-XL, GPT-J, and Llama3 models. The experiments are performed on 2000 edited samples, with an editing batch size of 100 (batch size refers to the number of samples edited simultaneously during each round of sequential editing), and evaluated on the Counterfact and ZsRE datasets. The evaluation results, using various metrics and across all datasets, are summarized in Table 1. From this table, we can observe that:

- **Observation 1: ECE outperforms other baseline methods in almost all critical metrics in the sequential editing task.** ECE demonstrates notable improvements compared to baseline methods across both datasets and models, achieving significant gains across all metrics. For instance, on the Llama3 (8B) model

Table 1: Comparison of ECE with existing methods on the sequential model editing task. The bold represents the best results from our methods and the underline indicates the best results of baselines. *Eff., Gen., Spe., Flu.* and *Consis.* denote Efficacy, Generalization, Specificity, Fluency and Consistency, respectively.

| Model | Method | Counterfact | | | | | ZsRE | | |
|---|---|---|---|---|---|---|---|---|---|
| | | Eff.↑ | Gen.↑ | Spe.↑ | Flu.↑ | Consis.↑ | Eff.↑ | Gen.↑ | Spe.↑ |
| | Pre-edited | 7.85±0.26 | 10.58±0.26 | 89.48±0.18 | 635.23±0.11 | 24.14±0.08 | 36.99±0.30 | 36.34±0.30 | 31.89±0.22 |
| Llama3 | FT-L | 83.33±0.37 | 67.79±0.40 | 46.63±0.37 | 233.72±0.22 | 8.77±0.05 | 30.48±0.26 | 30.22±0.32 | 15.49±0.17 |
| | FT-W | 61.23±0.38 | 62.40±0.24 | 47.05±0.41 | 492.34±0.23 | 3.57±0.03 | 32.08±0.35 | 31.43±0.23 | 14.72±0.16 |
| | MEND | 63.24±0.31 | 61.17±0.36 | 45.37±0.38 | 372.16±0.80 | 4.21±0.05 | 0.91±0.05 | 1.09±0.05 | 0.53±0.02 |
| | ROME | 64.40±0.47 | 61.42±0.42 | 49.44±0.38 | 449.06±0.26 | 3.31±0.02 | 2.01±0.07 | 1.80±0.07 | 0.69±0.03 |
| | MEMIT | 65.65±0.47 | 64.65±0.42 | 51.56±0.38 | 437.43±1.67 | 6.58±0.11 | 34.62±0.36 | 31.28±0.34 | 18.49±0.19 |
| | SERAC | 67.78±0.29 | 60.98±0.31 | 45.26±0.21 | 384.49±0.73 | 15.71±0.03 | 1.24±0.05 | 1.03±0.06 | 0.56±0.02 |
| | Ours (AS) | 92.90±0.10 | 82.85±0.27 | 80.93±0.20 | 628.32±0.14 | 31.62±0.11 | 89.29±0.14 | 83.25±0.25 | 30.03±0.23 |
| | Ours (WI) | **99.60±0.16** | **90.65±0.25** | 87.24±0.19 | 629.37±0.16 | **31.63±0.11** | **95.34±0.12** | **90.29±0.20** | **33.04±0.23** |
| | Ours (RS) | 97.50±0.15 | 85.25±0.31 | **87.93±0.19** | **631.09±0.13** | 31.00±0.10 | 93.81±0.15 | 88.38±0.22 | 32.92±0.23 |
| | Pre-edited | 22.23±0.73 | 24.34±0.62 | 78.53±0.33 | 626.64±0.31 | 31.88±0.20 | 22.19±0.24 | 31.30±0.27 | 24.15±0.32 |
| GPT2-XL | FT-L | 63.55±0.48 | 42.20±0.41 | 57.06±0.30 | 519.35±0.27 | 10.56±0.05 | 37.11±0.39 | 33.30±0.37 | 10.36±0.17 |
| | FT-W | 42.70±0.49 | 35.93±0.40 | 63.06±0.31 | 565.96±0.23 | 13.03±0.06 | 24.97±0.32 | 22.40±0.30 | 12.73±0.18 |
| | MEND | 50.80±0.50 | 50.80±0.48 | 49.20±0.51 | 407.21±0.08 | 1.01±0.00 | 0.00±0.00 | 0.00±0.00 | 0.00±0.00 |
| | ROME | 54.60±0.48 | 51.18±0.40 | 52.68±0.33 | 366.13±1.40 | 0.72±0.02 | 47.50±0.43 | 43.56±0.42 | 14.27±0.19 |
| | MEMIT | 94.70±0.22 | 85.82±0.28 | 60.50±0.32 | 477.26±0.54 | 22.72±0.15 | 79.17±0.32 | 71.44±0.36 | 26.42±0.25 |
| | SERAC | 51.50±0.44 | 50.04±0.42 | 52.13±0.47 | 418.12±0.76 | 1.55±0.02 | 38.58±0.36 | 41.49±0.46 | 13.78±0.21 |
| | Ours (AS) | 95.90±0.14 | 85.90±0.29 | 72.80±0.28 | 607.33±0.33 | **38.75±0.12** | 83.6±0.25 | 72.98±0.39 | **26.72±0.22** |
| | Ours (WI) | 96.20±0.19 | **89.10±0.31** | 78.44±0.27 | 625.74±0.13 | 32.84±0.10 | 86.71±0.39 | **79.33±0.33** | 26.12±0.25 |
| | Ours (RS) | **98.60±0.18** | 88.30±0.34 | 77.65±0.26 | 622.22±0.17 | 33.84±0.10 | **88.46±0.39** | 78.17±0.39 | 25.64±0.28 |
| | Pre-edited | 16.22±0.31 | 18.56±0.45 | 83.11±0.13 | 621.81±0.67 | 29.74±0.51 | 26.32±037 | 25.79±0.25 | 27.42±0.53 |
| GPT-J | FT-L | 92.15±0.27 | 72.38±0.38 | 43.35±0.37 | 297.92±0.77 | 6.65±0.10 | 72.37±0.29 | 68.91±0.32 | 19.66±0.23 |
| | FT-W | 48.35±0.49 | 31.42±0.39 | 68.71±0.28 | 587.20±0.23 | 29.41±0.09 | 39.81±0.36 | 32.55±0.33 | 27.76±0.26 |
| | MEND | 46.15±0.50 | 46.22±0.51 | 53.90±0.48 | 242.41±0.41 | 3.94±0.03 | 0.71±0.04 | 0.71±0.04 | 0.52±0.03 |
| | ROME | 57.50±0.48 | 54.20±0.40 | 52.05±0.31 | 589.28±0.08 | 3.22±0.02 | 56.42±0.42 | 54.65±0.42 | 9.86±0.16 |
| | MEMIT | 98.55±0.11 | 95.50±0.16 | 63.64±0.31 | 546.28±0.88 | 34.89±0.15 | 94.91±0.16 | 90.22±0.23 | 30.39±0.27 |
| | SERAC | 55.88±0.36 | 51.39±0.53 | 53.78±0.39 | 390.21±0.49 | 4.36±0.03 | 49.48±0.37 | 1.59±0.03 | 8.84±0.18 |
| | Ours (AS) | 98.82±0.09 | 95.73±0.23 | 74.25±0.26 | 618.5±0.23 | 42.22±0.13 | 96.20±0.15 | 93.35±0.25 | 27.19±0.21 |
| | Ours (WI) | **100.00±0.00** | 96.35±0.15 | 79.41±0.26 | **619.82±0.17** | 42.34±0.13 | **99.74±0.03** | **96.88±0.14** | 28.49±0.26 |
| | Ours (RS) | **100.00±0.00** | 96.45±0.14 | 79.99±0.25 | 619.38±0.17 | 41.10±0.13 | 97.28±0.13 | 94.99±0.21 | **28.86±0.24** |

with Counterfact dataset, ECE exhibits an average improvement of approximately 56.39% across the editing success rate containing efficacy, generalization, and specificity. On the ZsRE dataset, ECE's performance is even more remarkable, achieving multiple-fold improvements across all three models.

- **Observation 2: Three approaches' performances are evenly high in different situations.** The three methods within ECE—AS, WI, and RS—demonstrate consistently high performance across multiple models and datasets, reflecting the robustness and adaptability of each approach. For all tested configurations including Llama3, GPT2-XL, and GPT-J, three approaches, particularly WI and RS methods, achieve outstanding results. Notably, in task on

GPT-J model, both WI and RS approaches reach 100 % in efficacy metrics. While each explainer shows slight variations in specific metrics, they consistently maintain high fluency and specificity, suggesting balanced strengths for sequential batch editing tasks across varied model architectures.

## 4.3 Impact of Parameter (RQ2)

As the model undergoes successive modifications with editing tasks, sequential model editing methods face two inherent challenges: **model forgetting** and **model failure**. Model forgetting occurs when cumulative parameter changes from successive edits erode

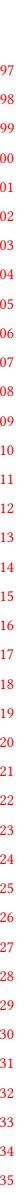

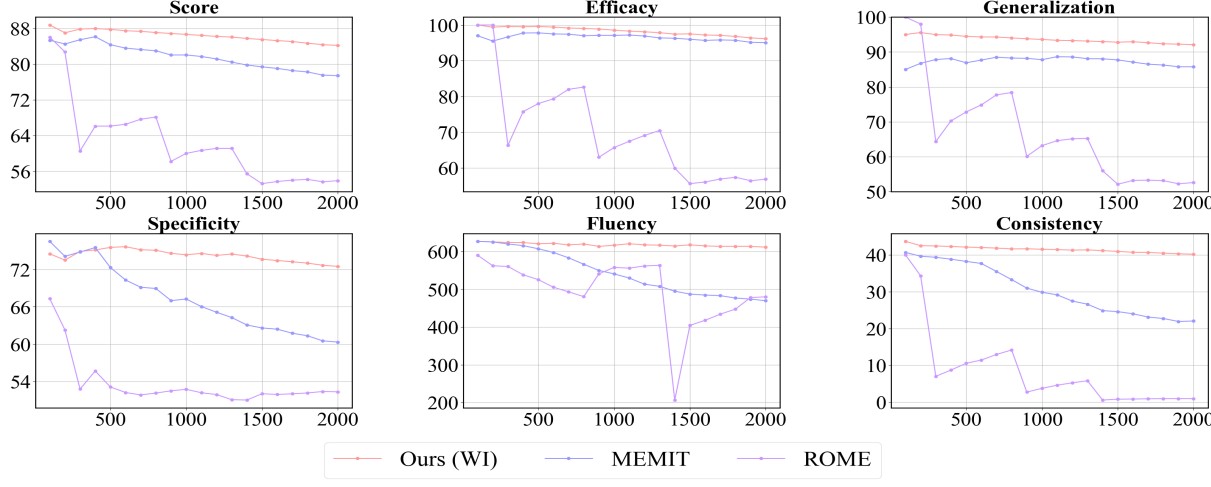

**Figure 3: Editing performance of ECE and baselines with different numbers of edits (batch size 100) evaluated on Llama3 model and Counterfact dataset. Score is the harmonic mean of Efficacy, Generalization, and Specificity.**

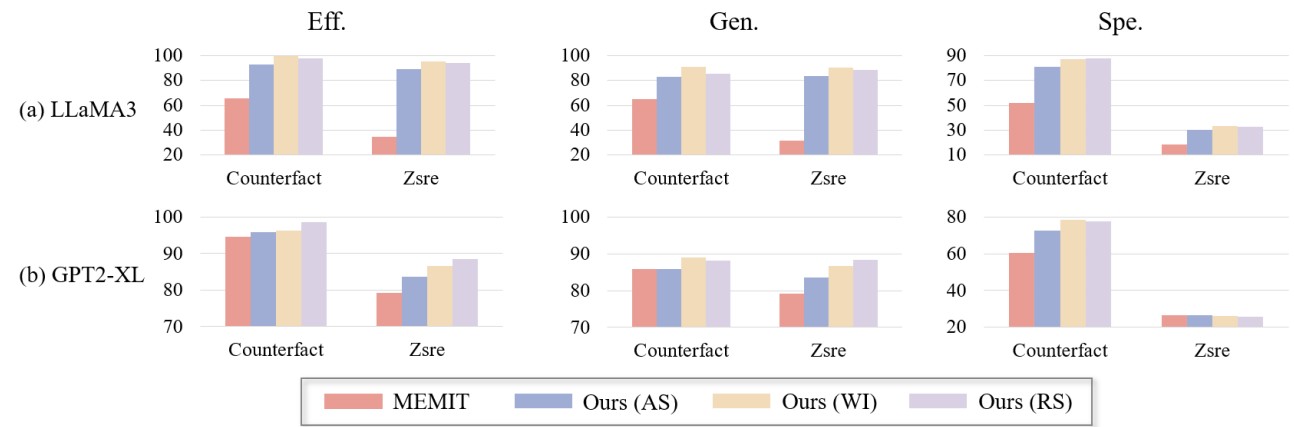

**Figure 4: Editing performance of MEMIT and ECE with 2000 edits in sequential editing, evaluated on the Counterfact and ZsRE dataset, on (a) Llama3 model and (b) GPT2-XL model.**

previously modified knowledge, resulting in a decline in performance and stability over time [13, 23]. Meanwhile, model failure refers to the progressive loss of the model's ability to generate coherent responses as edits accumulate, potentially leading to model collapse, where the output becomes repetitive or nonsensical [21, 22]. To explore these effects, we examine the influence of two key parameters **number of edits** and **batch size** on the sequential model editing process. Specifically, we analyze how the number of edits impact the performance of ECE compared to MEMIT and ROME on Llama3 and Counterfact dataset at 3. In Appendix 6, we presents how edit frequency influence model stability.

- **Observation 3: ECE maintains stable performance across all metrics as the number of edited samples increases.** As illustrated in Figure 3, ECE shows resilience against model failure and forgetting as the number of editing rounds grows. In contrast, both ROME and MEMIT experience considerable performance declines, particularly in Specificity, Fluency, and Consistency,. As

more samples are edited, ROME and MEMIT increasingly fail to uphold model integrity and impair model's original capabilities.

- **Observation 4: ECE consistently outperforms across a range of batch sizes in sequential editing tasks.** From 6 in E we can see that MEMIT's performance declines markedly as batch size decreases and the number of editing rounds increases. This effect is especially evident when the batch size is reduced to 10, showing a notable drop in editing effectiveness across all metrics. In comparison, ECE demonstrates stable performance across these metrics, regardless of batch size.

### 4.4 Time overhead Comparison (RQ3)

To evaluate the efficiency of our approach in sequential knowledge editing tasks, we conducted tests across three model architectures, benchmarking our method against established baselines. The evaluation involved a continuous editing scenario with a total of 2,000 edits and a batch size of 100. These numbers shown in Table 2

| Method | GPT2-XL | GPT-J | Llama3 |
|--------|---------|-------|--------|
| FT-L | 191.42s | 303.26s | 451.23s |
| FT-W | 157.44s | 263.74s | 374.35s |
| MEND | 26.79s | 49.16s | 67.85s |
| ROME | 422.37s | 764.82s | 914.63s |
| MEMIT | 222.51s | 334.74s | 484.14s |
| SERAC | 384.91s | 634.74s | 834.56s |
| Ours (AS) | 119.39s | 187.44s | 216.87s |
| Ours (WI) | 104.87.94s | 178.23s | 214.19s |
| Ours (RS) | 148.47.97s | 199.32s | 231.64s |

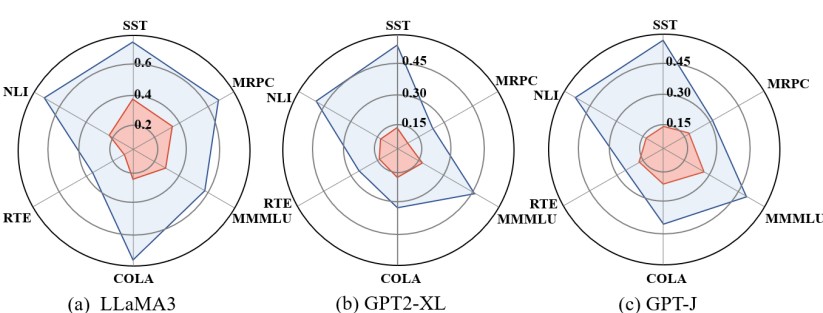

(a) LLaMA3     (b) GPT2-XL     (c) GPT-J

**Table 2: Times per batch for various methods evaluated on ZsRE dataset with different models.**

**Figure 5: Comparison of general capabilities for MEMIT and ECE (WI) with 2000 edits on (a) Llama3 model, (b) GPT2-XL model, and (c) GPT-J model.**

represent the average time of the whole editing process conducting at the first time. This means they could reflect the results of editing efficiency including getting expected output through gradient descent and further techniques.

- **Observation 5: Our methods consistently maintained superior efficiency, with the WI method being the fastest.** Our method demonstrated significant improvements in editing speed, surpassing nearly all baseline methods. Among our approaches, WI performs the best, followed closely by AS and RS. Although MEND displayed the shortest editing times, its low effectiveness on the ZsRE dataset limits its applicability, making it an unsuitable comparison. Overall, combined with results of editing performance, ECE achieves an noteworthy improvement in efficiency by acceleration approaches.

## 4.5 General Ability Test (RQ4)

To evaluate the impact of model editing on the general capabilities of large language models (LLMs), we have selected six natural language tasks from the General Language Understanding Evaluation (GLUE) benchmark [57], a public leaderboard for tracking performance with respect to a wide range of linguistic phenomena found in natural language. The chosen downstream tasks are as follows: (1) **SST (Stanford Sentiment Treebank)** [52], which involves classifying individual sentences based on sentiment in movie reviews. (2) **MRPC (Microsoft Research Paraphrase Corpus)** [14], a task focused on text matching to assess semantic similarity. (3) **MMLU (Massive Multi-task Language Understanding)** [26], which evaluates language models on multi-task accuracy. (4) **CoLA (Corpus of Linguistic Acceptability)** [61], a single-sentence classification task drawn from linguistic theory literature. (5) **RTE (Recognizing Textual Entailment)** [6], a natural language inference task to determine whether a given premise entails a hypothesis. (6) **NLI (Natural Language Inference)** [62], which requires the model to identify logical relationships between pairs of sentences. Case studies further illustrate the text generation effects of different editing methods, with detailed results provided in Appendix F.

We conduct evaluations on Llama3 (8B), GPT2-XL (1.5B), and GPT-J (6B) based on sequential editing settings with 2000 edits. From Figure 5 we can observe that:

- **Observation 6: ECE consistently maintains the general capabilities of the LLM during sequential editing without incurring model failure.** As the number of knowledge edits grows, ECE maintains performance levels comparable to those of the unedited LLMs, showing no negative impact on the model's core general capabilities. In contrast, both ROME and MEMIT have poor performance in different general capabilities, suggesting that the model has already suffered significant degradation. As shown in module (a), ECE achieves optimal performance across all generalization metrics in Llama3 model.

## 5 Limitation and Discussion

While ECE demonstrates significant improvements in both explainability and efficiency for sequential model editing tasks, there are still several limitations to our study. First, our evaluations are primarily focused on a few common and mainstream language models, such as GPT2-XL, GPT-J, and Llama3, which represent widely used architectures. Moreover, the experiments are currently based on existing datasets and specific environmental configurations. We have yet to test ECE on larger and more complicated datasets, which could present new challenges in terms of computational requirements and scalability. Looking ahead, we are committed to exploring more diverse techniques to further enhance the explainability and improve the overall efficiency and robustness of sequential editing, adapting it to a broader range of applications and advancing its capabilities for real-world deployment.

## 6 Conclusion

In summary, we presented Explainable and Efficient Sequential Editing (ECE), a method that addresses key limitations in the two-stage knowledge editing process for Large Language Models (LLMs). ECE enhances Stage 1 by adaptively identifying critical layers and neurons, leveraging model explainability for targeted updates. In Stage 2, ECE clusters similar keys to enable batch optimization, significantly reducing computational costs. Experimental results across different evaluation metrics and datasets demonstrate that ECE achieves superior editing performance with a substantial increase in efficiency, showcasing its potential to make model editing both explainable and efficient for real-world applications.

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

## A Related Work

### A.1 Model Editing

Model editing has emerged as an essential research area focused on modifying the behavior of pre-trained large language models (LLMs) to integrate new knowledge or correct factual errors, all without the need for extensive retraining.

**Preserve Models' Parameters.** Methods in this category aim to preserve the pre-trained model's parameters by introducing new knowledge through external components or retrieval mechanisms, rather than altering the core model itself. IKE [72] leverages in-context learning to adjust model outputs based on retrieved demonstrations, thus avoiding any gradient-based updates. Similarly, systems like SERAC [40] keep the model's parameters unchanged and use a counterfactual model to make edits, isolating the editing process from the base model. T-Patcher [27] introduces an additional neuron for each specific output error, while CaliNet [15] injects neurons to handle multiple knowledge cases. MELO [66] dynamically activates LoRA blocks indexed within an internal vector database, allowing models to behave differently depending on the retrieved block. On the other hand, GRACE [24] maintains a codebook to store knowledge and updates sequentially without modifying the core model. Similarly, Larimar [13] extends the idea of preserving model parameters by enhancing LLMs with a distributed episodic memory, which serves as an external knowledge source. OneEdit [68] introduces a neural-symbolic system that integrates knowledge graphs with LLMs for collaborative knowledge editing.

**Modify Models' Parameters.** Methods that modify LLMs' parameters focus on directly updating the internal weights of the model to incorporate new knowledge. FT-W [73] fine-tunes specific layers of the model using regularization constraints to ensure minimal changes to unrelated knowledge. Knowledge Neurons (KN) [11] identifies crucial neurons that encode factual knowledge within the feed-forward networks (FFNs) of the model and updates them accordingly. Similarly, methods such as KE [9] and MEND [39] employ hypernetworks to predict the necessary weight updates for new knowledge, leveraging meta-learning approaches to minimize computational overhead. ROME [37] and MEMIT [38] allow for large-scale direct editing of LLMs by locating and modifying specific knowledge in certain layers of models like GPT. ROME utilizes causal mediation analysis to identify the layers where knowledge is stored and performs targeted updates in these areas. MEMIT extends this approach, enabling simultaneous edits across multiple factual associations by modifying key neurons in the feed-forward layers. Building upon MEMIT, PMET [35] introduces attention values into the editing process, further improving performance by refining the selection of critical neurons for editing. To improve the stability and performance of parameter modification approaches especially for sequential model editing tasks, PRUNE [36] constrains the maximum singular value of parameter changes to avoid model degradation, while RECT [21] retains parameters with minimal changes to ensure stability. Additionally, COMEBA-HK [34] introduces hook layers to define the scope of editing, ensuring that changes are confined to the appropriate regions of the model, thus supporting sequential editing while maintaining performance on non-edited knowledge.

### A.2 Model Explainability

Due to the high computational costs involved and the assertion that only a select subset of neurons plays a crucial role in decision-making, existing methods are commonly combined with ranking algorithms to streamline the process [4]. Based on the premise that models learning similar properties often exhibit shared neurons, these neurons are ranked by metrics such as correlation coefficients and learned parameter weights [5, 12]. The Summarize and Score (SASC) [51] pipeline generates natural language explanations for large language model modules by first identifying n-grams that strongly activate the module and then evaluating these explanations with synthetic data to assess their relevance. The weight banding [45] studies weights that connect neurons, seeking to develop algorithms that reveal underlying logical structures. NAM [3] establish a complete attribution pipeline with adding the direct contributions through residual stream.

## B Detail of preliminary

Problem settings for model editing typically fall into four categories [65, 69]: single editing, batch editing, sequential editing, and sequential batch editing. In this work, we talks about the most complex type of editing.

(1) **Single Editing** assesses model performance after a single knowledge update:

$$\theta' \leftarrow \arg\min_{\theta} \left( \|f_\theta(x_i^e) - y_i^e\| \right) \tag{16}$$

(2) **Batch Editing** assesses model performance when multiple knowledge pieces are modified simultaneously ($n \leq N$ represents the batch size):

$$\theta' \leftarrow \arg\min_{\theta} \left( \sum_{i=1}^{n} \|f_\theta(x_i^e) - y_i^e\| \right) \tag{17}$$

(3) **Sequential Editing** requires that every single edit is executed successively and evaluation conducted only after all edits are completed []:

$$\theta' \leftarrow \arg\min_{\theta} \left( \sum_{i=1}^{N} \|f_\theta(x_i^e) - y_i^e\| \right) \tag{18}$$

(4) **Sequential Batch Editing** aims to perform edits in a sequential manner and in batches ($n$ represents the batch size, $S$ represents the sequential editing step):

$$\theta' \leftarrow \arg\min_{\theta} \left( \sum_{s=0}^{S} \sum_{i=s\times n}^{(s+1)\times n} \|f_\theta(x_i^e) - y_i^e\| \right) \tag{19}$$

## C  Details of Datasets and Evaluation Metrics

### C.1  Datasets

ZsRE [33] is a question answering (QA) dataset that employs questions generated via back-translation as equivalent neighboring prompts. In line with previous studies, natural questions are used as out-of-scope data to assess the locality aspect. Each ZsRE sample comprises a subject string and corresponding answers as the targets for evaluating editing success, along with rephrased questions for testing generalization and locality questions for assessing specificity.

Counterfact [29] is a more challenging dataset that distinguishes between counterfactual and factual statements, initially yielding lower scores for Counterfact. It generates out-of-scope data by substituting the subject entity with similar entities that share the same predicate. The Counterfact dataset includes metrics similar to those in ZsRE to evaluate efficacy, generalization, and specificity. Additionally, Counterfact offers multiple generation prompts with equivalent meanings to the original prompt to assess generated text quality, with a specific focus on fluency and consistency.

### C.2  ZsRE Metrics

Following the previous work [37–39], this section defines each ZsRE metric given a LLM $f_\theta$, a knowledge fact prompt $(s_i, r_i)$, an edited target output $o_i$, and the model's original output $o_i^c$:

- **Efficacy**: The efficacy metric is computed as the average top-1 accuracy on the edited samples:

$$\mathbb{E}_i \left\{ o_i = \arg\max_o \mathbb{P}_{f_\theta}(o \mid (s_i, r_i)) \right\}. \quad (20)$$

- **Generalization**: Generalization assesses the model's ability to perform on alternative prompts equivalent to $(s_i, r_i)$, such as paraphrased variations $N((s_i, r_i))$. It is calculated as the average top-1 accuracy on these paraphrased forms:

$$\mathbb{E}_i \left\{ o_i = \arg\max_o \mathbb{P}_{f_\theta}(o \mid N((s_i, r_i))) \right\}. \quad (21)$$

- **Specificity**: Specificity ensures that the edits do not alter model predictions on samples that are unrelated to the edited cases $O(s_i, r_i)$. This is measured by the top-1 accuracy of the predictions that remain consistent:

$$\mathbb{E}_i \left\{ o_i^c = \arg\max_o \mathbb{P}_{f_\theta}(o \mid O((s_i, r_i))) \right\}. \quad (22)$$

### C.3  Counterfact Metrics

Following prior works [37, 38], each Counterfact metric is defined for a large language model $f_\theta$, with a knowledge prompt $(s_i, r_i)$, an edited target output $o_i$, and the model's original output $o_i^c$:

- **Efficacy (edit success)**: The ratio of cases where $o_i$ has a higher probability than $o_c^i$ for the prompt $(s_i, r_i)$:

$$\mathbb{E}_i \left[ \mathbb{P}_{f_\theta}[o_i \mid (s_i, r_i)] > \mathbb{P}_{f_\theta}[o_c^i \mid (s_i, r_i)] \right]. \quad (23)$$

- **Generalization (paraphrase success)**: The proportion of cases in which $o_i$ is more likely than $o_c^i$ for rephrased prompts $N((s_i, r_i))$:

$$\mathbb{E}_i \left[ \mathbb{P}_{f_\theta}[o_i \mid N((s_i, r_i))] > \mathbb{P}_{f_\theta}[o_c^i \mid N((s_i, r_i))] \right]. \quad (24)$$

- **Specificity (unaffected prompt success)**: The fraction of neighboring prompts $O((s_i, r_i))$, referring to semantically related subjects, where the model maintains a higher probability on the accurate fact:

$$\mathbb{E}_i \left[ \mathbb{P}_{f_\theta}[o_i \mid O((s_i, r_i))] > \mathbb{P}_{f_\theta}[o_c^i \mid O((s_i, r_i))] \right]. \quad (25)$$

- **Fluency (repetition entropy)**: Measures output repetitiveness using entropy of n-gram distributions:

$$-\frac{2}{3} \sum_k g_2(k) \log_2 g_2(k) + \frac{4}{3} \sum_k g_3(k) \log_2 g_3(k), \quad (26)$$

where $g_n(\cdot)$ represents the n-gram frequency distribution.

- **Consistency (reference similarity)**: Consistency is evaluated by providing the model $f_\theta$ with a subject $s$ and then calculating the cosine similarity between the TF-IDF vectors of the generated text and a reference text (e.g., a Wikipedia entry) about $o$.

## D  Implementation details

### D.1  Implementation Details on GPT2-XL

The matrix $\lambda \mathbb{E} \left[ {}^T \right]$ is computed using 100,000 samples from Wikitext in fp32 precision, with the hyperparameter $\lambda$ set to 20,000. During the computation of $z_i$, we perform 20 epochs with a learning rate of 0.5. We set the threshold $p$ for neuron selection at 0.8. For other detailed parameters, we set clamp factor to 0.75, weight decay to 0.5, and kl factor to 0.0625. Those three parameters are set equally across three models.

### D.2  Implementation Details on GPT-J

The hyperparameter $\lambda$ is configured to 15,000. During the calculation of $z_i$, we conduct 25 iterations with a learning rate of 0.5, and the neuron selection threshold $p$ is set to 0.8.

### D.3  Implementation Details on Llama3 (8B)

We set the hyperparameter $\lambda$ to 15,000. In the calculation of $z_i$, we perform 25 iterations with a learning rate of 0.1, while maintaining the neuron selection threshold $p$ at 0.8.

### D.4  Additional Implementation Considerations

All experiments are executed on a single A100 (80GB) GPU for convenience, since fully running a single A40 (40GB) could handle almost every experiments. The language models loaded using HuggingFace Transformers [63]. To enhance both efficiency and resource management, we utilize the original model weights during the calculation of $z_i$. For practical storage optimization, we precompute $z_i$ values for all samples slated for editing and store these values, enabling direct access during editing without needing to retain the entire set of original model weights. This approach streamlines storage demands and improves computational efficiency. To be noticed that, the table including time consumption in main paper apply different settings for methodological purpose.

## E  Experiment

In this section, we present some supplemental information to the section 4 .

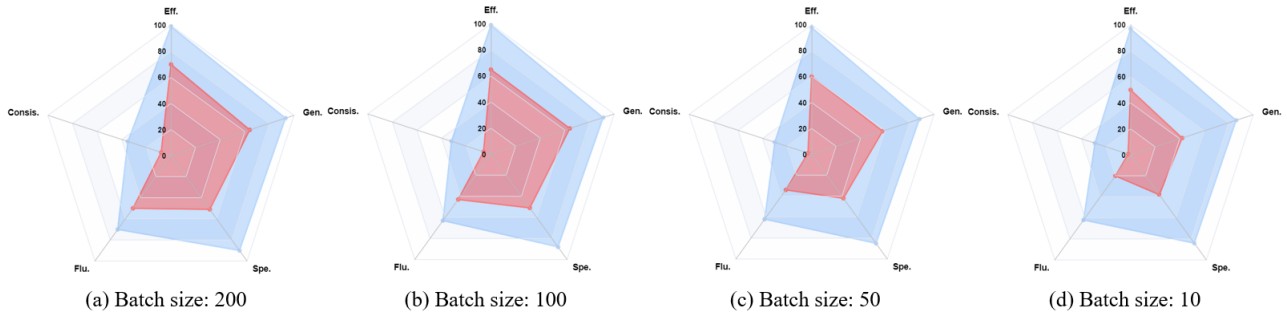

(a) Batch size: 200   (b) Batch size: 100   (c) Batch size: 50   (d) Batch size: 10

**Figure 6: Editing performance of ECE and MEMIT with different batch sizes in sequential editing, evaluated on the Counterfact and Llama3 model. The blue line and the red line represent ECE and MEMIT, respectively. Fluency's values are recomputed into the scale to align with others.**

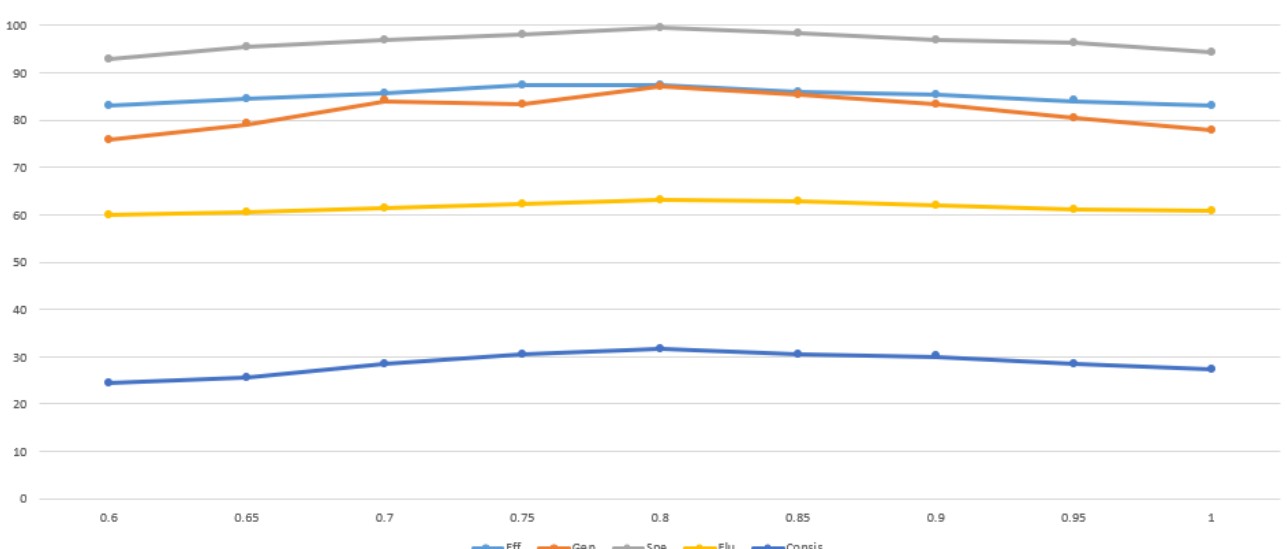

**Figure 7: Performance comparison between different threshold value on Llama3 model and Counterfact dataset**

Figure 6 corresponds to observation 4: ECE consistently outperforms across a range of batch sizes in sequential editing tasks. This helps to answer RQ2.

From figure 7, we determine to set the threshold value to 0.8 for achieving the best performance. We can see from 7, 0.8 is the highest point across different evaluation metrics, indicating the best option for experimental test.

# F Case Study

For a case study on generative capabilities, we examined an editing sample from the Counterfact dataset to compare the performance of ROME, MEMIT, and ECE after sequential editing. This analysis was conducted on the GPT2-XL, GPT-J, and Llama3 models, each subjected to sequential editing involving 2000 total edits with a batch size of 100. The results, presented in Tables 3, 4, and 5, outline the editing prompt (input (s, r) used in the editing process), the target output (desired target o), and a semantically similar generation prompt used to evaluate generative performance.

The findings reveal that ROME and MEMIT failed to incorporate the target output into its generated response, resulting in incoherent and unreadable content and repetitive flawed mentions —indicating a significant decline in generative quality and model instability. In contrast, our approach, ECE, not only achieved the edit successfully but also generated coherent, high-quality output, underscoring ECE's superior robustness and effectiveness in sequential editing tasks.

**Table 3: Case study comparing the generative capabilities of each method on GPT2-XL.**

| Case Study on GPT2-XL |
| --- |

| | |
| --- | --- |
| Editing Prompt: | Michel Denisot spoke the language |
| Target Output: | Russian |
| Generation Prompt: | *Michel Denisot's mother tongue is* |

| Generation Output |
| --- |

| | |
| --- | --- |
| **ROME:** | *Michel Denisot's mother tongue is* - Antarctica Antarctica Antarctica Antarctica Antarctica Antarctica Goat' HBO Antarctica Antarctica Antarctica Antarctica Antarctica save HBO Antarctica...Italy Antarctica Antarctica Antarctica Antarctica Antarctica Antarctica Antarctica Luigi fishes„ little HBO...s Munich Munich Munich Stefan Antarctica HBO and- fantasy Comcast City HBO City Luigi Sard, AntarcticaCity Lat Antarctica Goat HBO Antarctica City and Antarctica Antarctica City 0142 work Antarctic Goat' Antarctica Soda City, Antarctica City Antarctica Antarctica Antarctica RailCity original Antarctica City Antarctica Antarctica Antarctica Government Munich Antarctica and", |
| **MEMIT:** | *Michel Denisot's mother tongue is* Berlinannessanness Belgian Paris English German Paris German British London Berlin Czech Belgian Belgian politician Berlin musician politician Berlin politician French musician Dutch politician French politician German politician French politician Belgian politician politicsocr Belgian politician politician French politician Dutch politician Dutch politician Belgian politician politician Dutch politician politician Belgian politician politician politician French politician politician French politician politician politician French politician politician politician French politician politician French politician French politician journalist politician politician politician politician French politician French politician Belgian politician politician journalist politician Belgian politician politician politician", |
| **ECE:** | *Michel Denisot's mother tongue is* Russian. He is a dual Russian and French citizen. He was born in Moscow on 1 January 1987. His father is a French citizen, his mother is a Russian. He has a brother and sister. He has a sister in Moscow named Yulia (ŏ418ŏ43bŏ44cŏ44f ŏ423ŏ43bŏ44cŏ43cŏ430). He has a sister in Paris named Anastasia (ŏ410ŏ43dŏ430ŏ447ŏ430", |

**Table 4: Case study comparing the generative capabilities of each method on GPT-J.**

| Case Study on GPT-J | |
|---|---|
| Editing Prompt: | What does Heath Brothers play? They play |
| Target Output: | opera |
| Generation Prompt: | *Heath Brothers's greatest artistic work is* |
| **Generation Output** | |
| **ROME:** | *Heath Brothers's greatest artistic work is* Heath Brothers's greatest artistic work is google, AsiaTM Antarctica, global Global, Indian globally, Amazon Global, Antarctica.'s, is, sign,'s judgment international, UK,ultimate,ele prec, newly Derby, (constructive'sact wholes,uck secretaries,'s harmlesscel presumedactsecret's's's Area, Area, is,'s Area Global, secretaries,'s Area Global, barg South, sign'sactsecret's's Area Global, |
| **MEMIT:** | *Heath Brothers's greatest artistic work is* his expertise in real estate, and this expertise has been recognized by various organizations. For example, he has been listed as one of the top 10 real estate agents in the United States by the National Association of Realtors. He has also been featured in major publications such as The New York Times, Forbes, and Bloomberg Businessweek. estate is a highly competitive industry, and success requires a deep understanding of the market, strong negotiation skills, and the ability to", |
| **ECE:** | *Heath Brothers's greatest artistic work is* the opera The Pirates of Penzance. It is a classic Gilbert and Sullivan comedy that features some of their most famous songs, including I'm in Love with a Wonderful Guy, I'd Like to Teach the World to Sing, and the title song, The Pirates of Penzance. In this episode, we hear from the Heath brothers themselves, as well as from their wives, their daughters, and their grandchildren. We also hear from the singers", " |

**Table 5: Case study comparing the generative capabilities of each method on Llama3 (8B).**

| Case Study on Llama3 |
|---|

| Editing Prompt: | Triumph TR8 was created by |
|---|---|
| Target Output: | Boeing |
| Generation Prompt: | *Triumph TR8, produced by* |

| Generation Output |
|---|

| ROME: | *Triumph TR8, produced by* Nissan Ford BMW, BMW BMW BMW BMW Nissan Nissan Nissan BMW BMW Ford Nissan BMW Nissan BMW BMW Ford Nissan Chevrolet Nissan Nissan BMW Nissan BMW Nissan Nissan Nissan Chevrolet Nissan Nissan BMW Nissan BMW Nissan BMW BMW Nissan Nissan Nissan BMW BMW Nissan Nissan Nissan Toyota BMW BMW Nissan Nissan Nissan BMW Nissan BMW Ford Nissan Nissan BMW Nissan BMW Nissan Nissan BMW Nissan BMW Nissan BMW Nissan Nissan BMW Nissan Nissan Ford Nissan BMW BMW Ford Ford BMW Nissan Nissan Nissan BMW Nissan BMW Nissan Nissan", |
|---|---|
| MEMIT: | *Triumph TR8, produced by* the production planning department, which is responsible for the entire process from design to delivery. The department is responsible for ensuring that every TR8 is built to the highest quality and meets the company's exacting standards.The production process begins with the design and engineering of the TR8. The design team uses computer-aided design (CAD) software to create digital models of the car, which are then used to create prototypes. The prototypes are tested", |
| ECE: | *Triumph TR8, produced by* Boeing, a leading aerospace and defense company. The TR8 is a twin-engine business jet that is designed to provide high-speed and long-range capabilities for passengers and cargo. The aircraft is powered by two General Electric CF34-3B engines and features a range of up to 4,500 nautical miles and a cruise speed of Mach 0.82. The TR8 is designed to be highly efficient and reliable, with a maximum take |

## G  Ethical Statement

Our proposed ECE method aims to improve the efficiency and explainability of sequential model editing, making it highly suitable for dynamic, real-world applications that require frequent updates to stored knowledge. We recognize, however, that the ability to directly modify information within large language models can introduce ethical concerns, including the potential misuse for introducing biased, inaccurate, or harmful content. To mitigate these risks, we recommend rigorous validation procedures and continual oversight throughout the editing process. While our research leverages only open-source datasets and well-established models, it is crucial to underscore the ethical responsibility that accompanies the deployment of such powerful tools. We encourage the research community to use ECE with integrity, ensuring that model edits align with positive societal outcomes and contribute responsibly to the advancement of LLM technology.

