# OpenReview forum: "Explainable and Efficient Editing for Large Language Models"
_ACM.org/TheWebConf/2025/Conference — WWW 2025 Poster_

### Official Review · Reviewer_GhdP · 2024-11-25

**Novelty:** 4
**Technical Quality:** 3

**Review:**

This paper introduces the attribution methods into the large model knowledge editing, and improves the two stages of knowledge editing respectively.

Pros:

1.	ECE is more personalized and focused than previous approaches.

2.	Knowledge Editing is a compelling field of research.

Cons:

1.	The top half of Figure 1 is difficult to understand and the colors used are so similar as to be indistinguishable.

2.	Figure 7 has a two-column picture, but a single-column caption.

3.	I am confused about the task output. In your case study, the output is many tokens but the label is only one token. How to optimize the model parameters using equation 2 in this case?

4.	In your case study, the outputs of the model all precisely contain labels, but what if the output of the model contains synonyms for the labels? How should I determine this case?

5.	I observe that you show in Table 2 that ECE takes less time per batch, but ECE needs to do a forward for all the data separately, which might also take some time.

6.	What does the 104.87.94s in Table 2 mean?

7.	As far as I know, llama3 is much stronger than GPT2, but from the results of the experiment, there are some cases where GPT2 works better than llama3. Does this indicate that the datasets used for the experiment is too simple?

**Questions:**

Please see cons above.

**Reviewer Confidence:**

3: The reviewer is confident but not certain that the evaluation is correct

**Scope:**

3: The work is somewhat relevant to the Web and to the track, and is of narrow interest to a sub-community

---

### Official Review · Reviewer_TS8i · 2024-11-28

**Novelty:** 4
**Technical Quality:** 2

**Review:**

The proposed work introduces ECE (Explainable and effiCient model Editing), a novel approach for model editing in large language models (LLMs). The method addresses limitations of current two-stage paradigms in knowledge editing, namely the independent identification of critical layers and the high computational cost associated with updating parameters. ECE incorporates explainability into the identification process and enables batch optimization to reduce computational overhead.

**Questions:**

Lack of Theoretical Foundation
While the method is well-explained in practice, there could be more theoretical grounding for why explainability-driven approaches are superior in this context. A deeper exploration of how knowledge is represented in the layers and why certain neurons are critical would enhance the scientific rigor.

Optimization Cost in Large Models
While the method achieves a speedup, the optimization cost in very large models with billions of parameters may still be prohibitive in certain contexts. Further investigation into how ECE scales with the increasing size of LLMs is necessary.

Limited Comparison with State-of-the-Art
Although the paper showcases superior performance in editing efficiency, a more detailed comparison with other state-of-the-art techniques could help better position ECE in the broader landscape of model editing methods.

**Reviewer Confidence:**

2: The reviewer is willing to defend the evaluation, but it is likely that the reviewer did not understand parts of the paper

**Scope:**

3: The work is somewhat relevant to the Web and to the track, and is of narrow interest to a sub-community

---

### Official Review · Reviewer_dcSB · 2024-12-01

**Novelty:** 6
**Technical Quality:** 6

**Review:**

This work proposes ECE, a method for model editing of large language models. Experiments demonstrate its performance gains and speedup.

Overall this is a solid work. Several minor comments are:

- The efficiency comparison in Table 2 is based on wall time (seconds). However, the exact wall time it takes to run a program vary based on machine hardware: I wonder if more machine-agnostic metrics like FLOPs might be employed here for calculation.

- I wonder if the authors thought about evaluating the ripple effect of knowledge editing that comes with this approach: is it better or worse than say ROME or MEMIT?

- Two of the three LLMs in the main experiments (GPT2-XL and GPT-J) are really weak and outdated, but given that there is llama3 this shouldn't be a huge concern.

- The related work is put in the appendix: I know that the authors are trying to make more space for results and analysis in the main paper, but this is perhaps not good practice.

**Questions:**

please see above

**Reviewer Confidence:**

3: The reviewer is confident but not certain that the evaluation is correct

**Scope:**

3: The work is somewhat relevant to the Web and to the track, and is of narrow interest to a sub-community

---

### Official Review · Reviewer_MPFq · 2024-12-02

**Novelty:** 6
**Technical Quality:** 6

**Review:**

- This paper proposes a efficienct and explainable LLM editing method with innovative methodology design and adequate experimental results.
- The pain points that the authors are trying to address, including explainability and efficiency, are some of the more critical topics in the current field of model editing

### Weaknesses
- More experiments on larger models (>=13B) could be conducted to present the effectiveness and efficiency of the method.
- More experiments and case studies about the explainability of the method could be presented in the main content. Now we only see some output cases in the Appendix. More cases to explain the internal mechanisms of the editing process could be added.
- No code and data is submited for reproducing the experiments.
- No related work is presented in the main content. Authors put them in the Appendix.

**Questions:**

- Is there any typos in Table2? I found some experimental data has two decimal points.

**Reviewer Confidence:**

3: The reviewer is confident but not certain that the evaluation is correct

**Scope:**

3: The work is somewhat relevant to the Web and to the track, and is of narrow interest to a sub-community